# A Standardized Method for Estimating the Functional Diversity of Soil Bacterial Community by Biolog® EcoPlates™ Assay—The Case Study of a Sustainable Olive Orchard

**Adriano Sofo [1,*]** and **Patrizia Ricciuti [2]**

1   Department of European and Mediterranean Cultures: Architecture, Environment and Cultural Heritage (DiCEM), University of Basilicata, Via San Rocco, 3–75100 Matera, Italy
2   Department of Soil, Plant and Food Sciences (DiSSPA), University of Bari 'Aldo Moro', Via Amendola, 165–70126 Bari, Italy; patrizia.ricciuti@uniba.it
*   Correspondence: adriano.sofo@unibas.it



**Featured Application:** **Soil bacteria are of paramount importance for determining soil quality and fertility but evaluating the microbiological status of the soils in a simple and reliable way is not easy. For doing this, specific plates are commercially available but, in order to have uniform results, a common procedure should be followed by everyone. We tried to fill this gap. In addition to agricultural production, our results are useful for mitigating soil pollution and climate change, where bacteria play a key role.**

**Abstract:** Biolog® EcoPlates™ (Biolog Inc., Hayward, CA, USA) were developed to analyse the functional diversity of bacterial communities by means of measuring their ability to oxidize carbon substrates. This technique has been successfully adopted for studying bacterial soil communities from different soil environments, polluted soils and soils subjected to various agronomic treatments. Unfortunately, Biolog® EcoPlates™ assay, especially working on soil, can be difficult to reproduce and hard to standardize due to the lack of detailed procedures and protocols. The main problems of this technique mainly regard soil preparation, bacterial inoculum densities and a correct definition of blank during the calculation of the diversity indices. On the basis of our previous research on agricultural soils, we here propose a standardized and accurate step-by-step method for estimating the functional diversity of a soil bacterial community by Biolog® EcoPlates™ assay. A case study of soils sampled in a Mediterranean olive orchard managed accordingly to sustainable/conservation practices was reported for justifying the standardized method here used. The results of this methodological paper could be important for correctly evaluating and comparing the microbiological fertility of soils managed by sustainable/conservation or conventional/non-conservation systems.

**Keywords:** Biolog®; community-level physiological profiling (CLPP); functional diversity indices; metabolic bacterial diversity; olive; soil fertility; soil quality

## 1. Introduction

Microorganisms are present in all ecosystems and due to their rapid responses to physical and chemical changes, they can be used as bioindicators of environmental quality. The Community-Level Physiological Profiling (CLPP) is a rapid and relatively inexpensive technique to relate microbial functional diversity over space and time to changes in the environment [1–4].

Biolog® EcoPlates™ (Biolog Inc., Hayward, CA, USA) were developed to analyse the functional diversity of bacterial communities by means of measuring their ability to oxidize carbon substrates. An EcoPlate is a 96-well microplate that contains 31 common carbon sources from altogether five compound groups—that is, carbohydrates, carboxylic and ketonic acids, amines and amides, amino acids and polymers—plus a blank well as a control, all these replicated thrice to control variation in inoculum densities. Each EcoPlate is filled with a dilution of one soil suspension, thus representing one soil sample. The utilization rates of carbon compounds in the wells are quantified spectrophotometrically by following the reduction of water-soluble colourless triphenyl tetrazolium chloride to purple triphenyl formazan. For the measurements of optical density (OD), two filters are used: (a) 590 nm (absorbance peak of tetrazolium) to evaluate colour development plus turbidity values and (b) 750 nm to measure turbidity values only. The turbidity of dilutions is due to clay and humic particles in soil colloidal suspension.

Every bacterial community has a characteristic reaction pattern with different OD values for different carbon compounds, called a 'metabolic fingerprint' [2,5]. While the inoculated bacterial density significantly affects the rate of colour development in the wells, on the other side the choice of a wrong inoculum can compromise the results of this techniques [5,6]. For this reason, it should be necessary to accurately choose the inoculum densities for different soil samples using a plate count culture-based method. Moreover, a correct definition of blank is essential for calculating the related diversity indices. Biolog® EcoPlates™ have been successfully adopted in our laboratories for studying bacterial soil communities from different soil environments, polluted soils and soils subjected to various agronomic treatments [6–9]. Unfortunately, the studies regarding soil as a matrix for bacterial communities often include a Biolog® analysis but the reagents and apparatus for soil preparation, dilution and incubation time used, standard deviations and number of replicates and precise formulas for calculations are often not reported, so it becomes difficult for other researchers to understand how to correctly use this method.

On this basis and taking into account our previous research on soils, we here propose for the first time a standardized and accurate step-by-step method for estimating the functional diversity of a soil bacterial community by Biolog® EcoPlates™ assay. We tried to adopt a clear terminology and parameters, explaining them in detail, in order to facilitate the calculation of the most relevant and used Biolog®-related indices of microbial functional diversity. Even with its limitations [10], the Biolog® EcoPlates™ method remains a quick and relatively simple and inexpensive technique for comparing microbial communities. The standardization of this technique could be essential for indicating differences in CLPPs between samples within a single experiment, for comparing different experiments, soil types and management systems or simply soils sampled at different times. The aim of this report is not describing the limitations of the method itself but proposing a detailed methodological procedure, easy to be followed and reproduced and based on previous data and experiments. A case study of soils of a Mediterranean olive orchard was reported for comparing the methods and justify the methodological protocol here proposed.

## 2. Materials and Methods

### 2.1. Methodology

#### 2.1.1. Culture-Based Plate Count Method

Plate count must be conducted under sterile conditions in a laminar-flow hood using single-use sterile plastic material and autoclaved solutions and glassware. The following materials are used: sterile flasks and tubes, sterile spatula, sterile pipettes, P1000 and P100 micropipettes with sterile tips, Petri dishes (size 90 mm, polystyrene, $\gamma$-irradiated), sterile hockey stick (disposable cell spreaders). The apparatus includes laminar flow hood, autoclave, incubator, water bath, magnetic stirrer, ultrasonic bath. The reagents are: Tryptic Soy Agar (TSA), cycloheximide (to inhibit fungal growth), 25% sterile

Ringer solution (NaCl 2.25 g $L^{-1}$, KCl 0.105 g $L^{-1}$, CaCl$_2$ 0.045 g $L^{-1}$ and NaHCO$_3$ 0.05 g $L^{-1}$), 1.8% (*w/v*) sterile sodium pyrophosphate (Na$_4$P$_2$O$_7$ • 10 H$_2$O) solution.

For making and plating, 3 g of TSA powder were added to 1 L of distilled water in a 2-L glass bottle. This solution was sterilized (autoclaved) at 121 °C for 20 min, cooled at 50 °C and then 100 μg cycloheximide mL$^{-1}$ were added mixing thoroughly. Finally, 20 mL of the solution were poured in each Petri dish.

For plate counting, in order to obtain a 0.18% (*w/v*) Na$_4$P$_2$O$_7$ • 10 H$_2$O final concentration, 4.5 mL of 1.8% sodium pyrophosphate and 40.5 mL of 25% Ringer solution were added to 5 g (dry weight equivalent) of fresh soil. The suspension was sonicated for 2 min and soil particles were allowed to settle at 4 °C for 15 min. Then, ten-fold serial dilutions of the supernatant up to $10^{-7}$ in sterile Ringer solution were done, spreading a 100 μL-aliquot of each dilution onto a TSA plate (3-5 replicates for each dilution) and incubating at 28 °C for 72 h. For each sample the suited dilution to enumerate colony forming units (CFUs) for g of dried weight soil was chosen. Then, for microplate incubations, the dilution leading to ~$10^4$ CFUs mL$^{-1}$ solution was selected for microplate incubation.

### 2.1.2. Microplate Incubation

The following materials are used: multichannel pipet and sterile tips, sterile plastic multichannel reservoir, Biolog® EcoPlates™ (Biolog Inc., Hayward, CA, USA). The apparatus includes laminar flow hood, incubator, agitator and Biolog® Microplate Reader™ equipped with 750-nm and 590-nm filters.

For preparing the microplate, 10 mL of the dilution that was chosen in plate counting was poured into a sterile reservoir of an 8-channel pipet (be careful there are no bubbles in the dilution) and 120 μL of the dilution were inoculated into each well of a microplate. Not the same dilutions of the counting assay were used but new fresh dilutions were prepared. Then, the microplate was placed in its bag to avoid desiccation and incubated at 25 °C in dark, continuously shaking at 50 rpm on an agitator with tilting platform to obtain a uniform distribution of triphenyl formazan. Finally, spectrophotometric readings at both 590 nm (OD$_{590}$) and 750 nm (OD$_{750}$) were taken at time 0 and at 12-h increments ($\pm 12_{Xh}$) up to a 144-h incubation.

In order to select the optimal incubation time for microplate analyses (as explained later in the Results and Discussion Section), it is recommended to follow this pattern for each incubation time (as an example, measuring times at 0 h and X h are given here):

- calculate a colour value for each substrate well *i* and the blank (water) well *b* for each incubation time by subtracting the OD$_{750}$ value from the OD$_{590}$ value:

$$0\ \text{h:} \quad i_{0h} = \text{OD}_{590} - \text{OD}_{750} \quad \text{and} \quad b_{0h} = \text{OD}_{590} - \text{OD}_{750}$$

$$X\ \text{h:} \quad i_{Xh} = \text{OD}_{590} - \text{OD}_{750} \quad \text{and} \quad b_{Xh} = \text{OD}_{590} - \text{OD}_{750}$$

- subtract the blank well OD reading from the OD value of each substrate well to obtain a blank-corrected value ($i_{bc}$) for each well:

$$0\ \text{h:} \quad i_{bc0\text{h}} = i_{0h} - b_{0h}$$

$$X\ \text{h:} \quad i_{bc X\text{h}} = i_{Xh} - b_{Xh}$$

- subtract the blank-corrected OD reading at time 0 from subsequent blank-corrected readings at $\pm 12_{Xh}$ to obtain colour development values ($c_i$) for each well for each incubation time: for example, $c_{iXh} = i_{bcXh} - i_{bc0h}$ and set negative values to 0
- calculate the average well colour development (*AWCD*) for all incubation times separately using the equation:

$$AWCD = \sum \frac{c_i}{93}$$

### 2.1.3. Utilizing the AWCD and $c_i$ Values

The *AWCD* calculated above is an estimate of the total capacity of a bacterial community to use different carbon compounds. Using the $c_i$ values of the chosen incubation time, it is possible to further calculate indices of bacterial functional diversity [2,11], such as:

(a)  Richness (S), which is the number of utilized carbon substrates, using an OD value of 0.250 as threshold for positive response
(b)  Shannon's diversity index (H'), which is related to the number of carbon substrates the bacterial community is able to degrade

$$H' = -\sum p_i \left( \ln p_i \right)$$

where $p_i$ is $c_i$ divided by the sum of all the $c_i$ values.

(c)  and Shannon's evenness index (E), which particularly focuses on the evenness of ci values across all utilized substrates

$$E = \frac{H'}{\ln S}$$

For a more detailed analysis, the carbon substrates can eventually be divided into eight classes of compounds (polysaccharides and complex compounds, cellulose, hemicellulose, chitin, phosphorylated compounds, organic acids, amino acids and biogenic amines) and the *AWCD* and diversity indices calculated for each group separately (see examples in References [4,5] and in Figure 1).

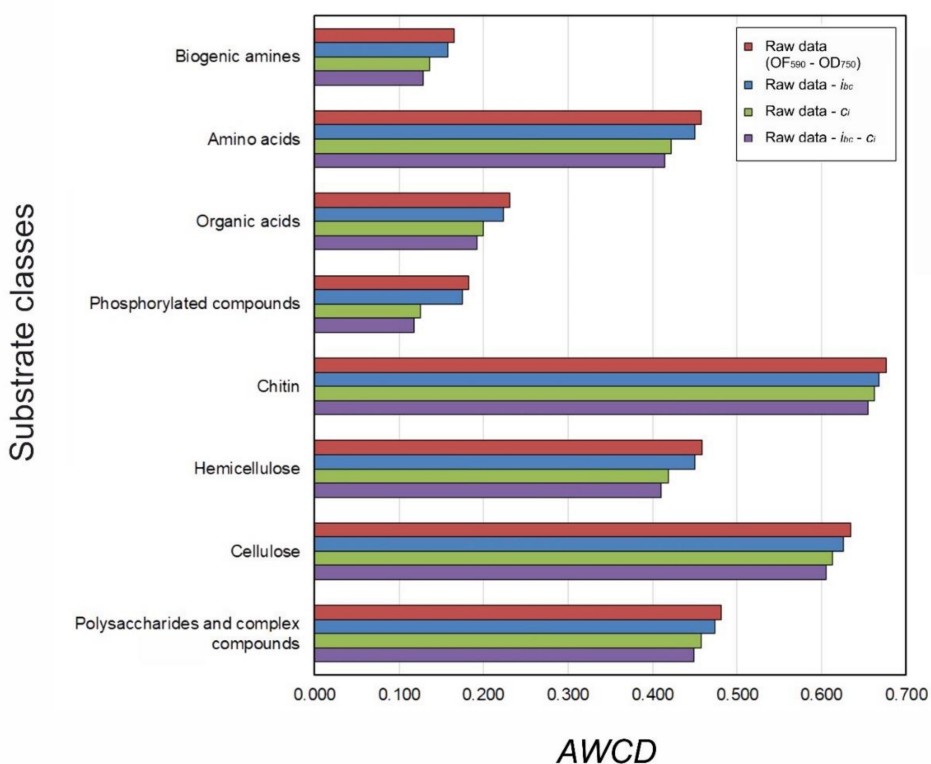

**Figure 1.** Mean values (*n* = 5) of average well colour development (*AWCD*) measured after a 96-h incubation in dilutions (~$10^4$ CFUs $mL^{-1}$) of soils from the experimental olive orchard. The values were calculated subtracting from the raw data ($OD_{590} - OD_{750}$) the blank well optical density (OD) reading from the OD value of each substrate well ($i_{bc}$) for each well and successively subtracting the blank-corrected OD reading at time 0 from subsequent blank-corrected readings at $\pm12_{Xh}$ ($c_i$) (in purple). The same methodology was carried out without subtracting the blank well OD reading (in green), without subtracting the blank-corrected OD reading at time 0 (in blue) or without subtracting both (raw data in red).

*2.2. Case Study and Methods Comparison*

Experimental Field and Soil Sampling

The trial was done in a 1-ha olive orchard (*Olea europaea* L., cv. 'Maiatica'; 70-year-old plants with a distance of 8 × 8 m; NE orientation) located in Ferrandina (Southern Italy, Basilicata region; N 40°29′; E 16°28′). The area has a semi-arid climate, with an annual rainfall of 560 mm (mean 1995–2018), concentrated mostly in the winter and a mean annual temperature of 16.3 °C (mean 1995–2018). The soil is a sandy loam, a Haplic Calcisol, according to the World Reference Base for Soil Resources, with a mean bulk density of 1.30 g cm$^{-3}$ and sediment as parental material.

The orchard was managed accordingly to sustainable/conservation agricultural practices, namely: (a) minimum tillage and cover crop application (30 kg ha$^{-1}$ of *Trifolium subterraneum* seeds and spontaneous grass); (b) guided fertilization based on plant nutrient demand evaluated by leaf mineral analyses and on soil measured nitrogen levels; (c) compost amendment (15 t ha$^{-1}$ fresh weight); (d) incorporation of cover crop and pruning residues into the soil (light harrowing at a depth of 10 cm carried out in Autumn); (e) guided drip irrigation based on crop evapotranspiration (3 drip emitters per plant along the tree lines with a capacity of 4 L h$^{-1}$ each); (f) pruning aimed to vegetative-productive equilibrium of plants (winter pruning based on the selection of shoots with a high number of floral buds and on a better light interception in the canopy).

In October 2018, soil sampling was done in the undisturbed inter-row area. Soil sub-samples were picked in 10 points over a 1-m radius area around each olive tree from the topsoil layer (0–10 cm) for bacterial communities' analysis by Biolog® assay. The 10 sub-samples were pooled on site to constitute a composite soil sample of about 1 kg. Five composite samples (*n* = 5), each composed of 10 different sub-samples, were prepared, in order minimize spatial variability. After removing visible crop residues, roots and pebbles with sterile tweezers and slightly mixing and homogenizing with a sterile spatula, the soil composite samples were immediately stored in sterilized plastic bags at 4 °C and subsequently analysed within 5 days using the procedure described in the previous paragraph.

## 3. Results and Discussion

*3.1. Influence of Bacterial Inoculum Densities*

For the best discrimination of bacterial communities, it is of fundamental importance to choose the shortest incubation time at which *AWCD* reaches a peak value before the following constant phase. Figure 2A shows the *AWCD* values measured every 12 h during a 144-h incubation from different soil dilutions. In our case the dilutions examined were 10$^{-5}$, corresponding to ~10$^4$ colony forming units (CFUs) mL$^{-1}$ measured by plate counting, 10$^{-4}$ and 10$^{-6}$. It appears that the dilution with 10$^{-6}$ dilution contained a relatively low number of bacteria (~10$^3$ CFUs mL$^{-1}$) causing an increasing *AWCD* trend over time, without reaching a constant phase before 144 h and high variability among replicates, as showed by the high values of standard deviation (Figure 2A). On the other side, the 10$^{-4}$ dilution likely had too many bacteria (10$^5$ CFUs mL$^{-1}$) and *AWCD* reached a peak after only 48 h, followed by a slight decline likely due to substrates or triphenyl formazan degradation, without a clear constant phase (Figure 2A). Instead, the dilution of 10$^{-5}$ (~10$^4$ CFUs mL$^{-1}$) reached a constant phase at 96 h and so it was chosen as the time for measuring *AWCD* values (Figure 2A). Similar trends were observed both for *H*′ (Figure 2B) and *E* values (Figure 2C), that had a constant phase starting from 96 h in the dilution of 10$^{-5}$ (~10$^4$ CFUs mL$^{-1}$), with low values of standard deviation from 96 to 120 h of incubation (Figure 2B,C). The common pattern of the graphs of Figure 2 likely depends on the fact that the calculation of *AWCD*, *H*′ and *E* are all based on the values of $c_i$, as previously explained.

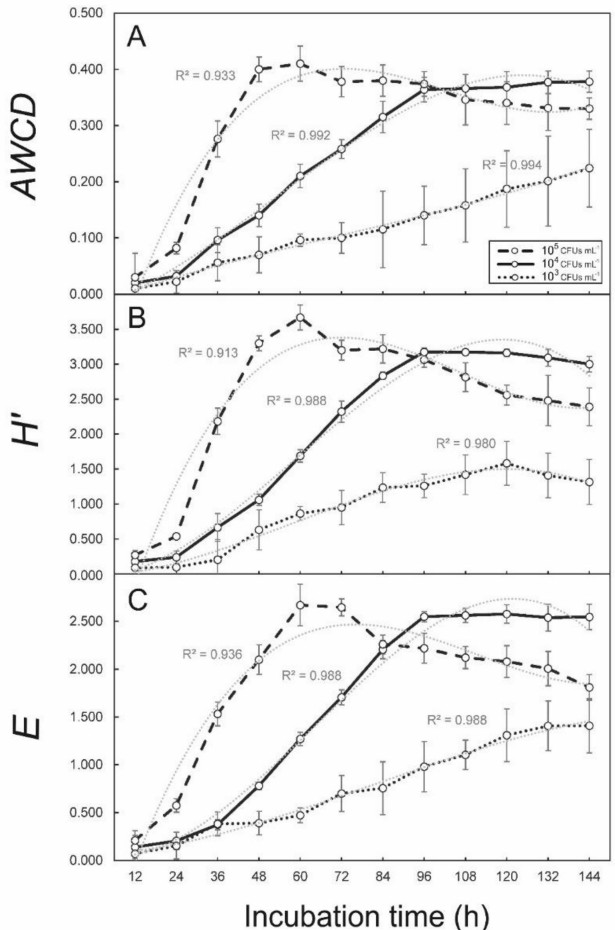

**Figure 2.** Mean values (*n* = 5) ± SD of (**A**) *AWCD*, (**B**) *H′* and (**C**) *E* measured every 12 h during a 144-h incubation in different dilutions (~$10^5$ CFUs mL$^{-1}$, dashed line; ~$10^4$ CFUs mL$^{-1}$, continuous line; ~$10^3$ CFUs mL$^{-1}$, dotted line) of soils from the experimental olive orchard. $Na_4P_2O_7 \bullet 10\ H_2O$ concentration was 1.8% (*w/v*). The $R^2$ values and the respective trend lines (dotted lines in grey) were calculated on the basis of a polynomial function of degree three.

Comparing our results with those of other studies [10–16] using our protocol, we managed to obtain more stable growth curves for *AWCD*, particularly in the constant phase and relatively low values of SD (Figure 2A). Another important remark is that the inoculum could be relativized simply resuspending soil mass, in order to not to artificially homogenize differences that are intended to be detected among the different soil samples of the same experiment [17–20]. In order to avoid this possible artefact, in our case, the same dilution (leading to ~$10^4$ CFUs mL$^{-1}$) was chosen for all the samples analysed in the same experiment.

### 3.2. Correct Calculation of Blank

The *AWCD* values measured on the $10^{-5}$ dilutions at 96 h were calculated subtracting from the raw data ($OD_{590} - OD_{750}$) the blank well OD reading from the OD value of each substrate well ($i_{bc}$) for each well and successively subtracting the blank-corrected OD reading at time 0 from subsequent blank-corrected readings at $\pm12_{Xh}$ ($c_i$) for both 0-h and 96-h incubation times. The same methodology was carried out (a) without subtracting the blank well OD reading, (b) without subtracting the blank-corrected OD reading at time 0 or (c) without subtracting both. The results, depicted in Figure 1, were quite different on the basis of the type of calculation done, particularly for some classes of carbon substrates (e.g., phosphorylated compounds), showing that it is necessary to apply the

sequential subtractions and that non-specific OD values must be taken into account for obtaining reliable *AWCD* data.

### 3.3. Influence of Soil Preparation

On the same soil samples ($10^{-5}$ dilution, reading at 96 h, subtraction of both blank well and blank-corrected OD readings) and on the basis of previous experiments [7,9], eight concentrations of $Na_4P_2O_7 \bullet 10 H_2O$ ranging from 0.0% to 3.0% (*w/v*) (0, 0.6, 1.0, 1.4, 1.8, 2.2, 2.6 and 3.0%) were used for finding the optimal concentration to be used. Due to its high number of electric charges once in solution, it is known that this compound has the ability to disperse charged soil colloids. This is particularly important for soil microbiological assays, as bacteria are not only present on the surface of soil aggregates but also inside the aggregates, that is, on the micropore and macropore walls and in their internal spaces [6]. Figure 3 shows that with more than 1.8% (*w/v*) $Na_4P_2O_7$ soils did not show an increase in *AWCD*, *H'* and *E*, probably because the maximum possible number of bacteria was extracted from soil. Of course, this value could vary for different soil types but we can suggest that a concentration of around 2.0% (*w/v*) is necessary for not underestimating *AWCD* values. The use of the appropriate concentration of $Na_4P_2O_7$ allows to overcome the technical problem in extracting the highest number of microorganisms from soil samples, that was one of the main remarks of this technique highlighted by Preston-Mafham et al. [10].

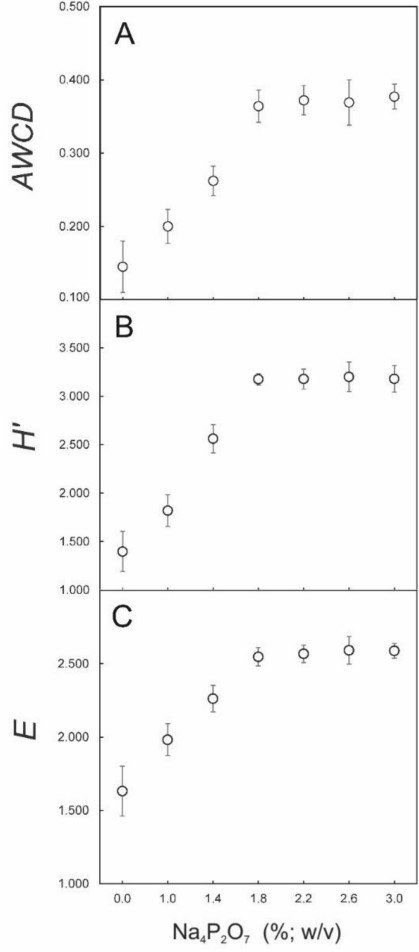

**Figure 3.** Mean values (*n* = 5) ± SD of (**A**) *AWCD*, (**B**) *H' and* (**C**) *E* measured a 96-h incubation in dilutions (~$10^4$ CFUs $mL^{-1}$) of soils from the experimental olive orchard. The dilutions were obtained using different $Na_4P_2O_7 \bullet 10 H_2O$ concentrations.

## 4. Conclusions

A detailed and step-by-step description in the materials and methods sections of the scientific articles dealing with Biolog® is often lacking, making this method hardly reproducible. For this reason, we hope that the description of this method can help to solve the main issues related to the Biolog® EcoPlates™ assay: (1) soil preparation, dispersion and dilution (importance of using composite samples and $Na_4P_2O_7$); (2) the optimal bacterial density of the inoculum (~$10^4$ CFUs mL$^{-1}$); (3) a correct definition for blank ($c_{iXh} = i_{bcXh} - i_{bc0h}$); and (4) the use of unequivocal terminology and parameters in the calculation of diversity indices, as reported in the formulas. The previously-cited problems make this technique apparently difficult to reproduce and hard to standardize. This notwithstanding, we tried to demonstrate that, if rigorously executed, Biolog® EcoPlates™ assay can be a relatively simple and powerful technique, extremely useful for evaluating soil bacterial functional and metabolic diversity. The results of this methodological paper could be important for correctly evaluating and comparing the microbiological fertility of soils managed by sustainable/conservation or conventional/non-conservation systems.

**Author Contributions:** Conceptualization, A.S.; Methodology, P.R. and A.S.; Validation, P.R.; Data Curation, A.S.; Writing—Original Draft Preparation, A.S.; Writing—Review & Editing, P.R. and A.S.

**Funding:** This work was partly supported by an OECD Co-operative Research Programme grant: Biological Resource Management for Sustainable Agricultural Systems. Directorate: T AD/CRP; Contract: JA00091460.

**Conflicts of Interest:** The authors declare no conflict of interest.

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
