# Peer review of "A Standardized Method for Estimating the Functional Diversity of Soil Bacterial Community by Biolog® EcoPlatesTM Assay—The Case Study of a Sustainable Olive Orchard"

_applsci, doi:10.3390/app9194035_

Round 1

Reviewer 1 Report

This manuscript represents a methodological contribution to an usual problem in the technical management and data analysis associated with the implementation of the BIOLOG Ecoplate test. However, I consider that it is not useful for all the possible applications of this system so you must be more precise about the standardization that you propose and for which kind of research it will be useful. The article is presented concisely, but it requires adjustments to facilitate its understanding and justify the proposal.

You must reference this affirmation in lines 57-59: “However, the inoculated bacterial density significantly affects the rate of color development in the wells and therefore, inoculum densities have to be standardized for different soil samples using a plate count culture-based method”. It is important that you indicate for which type of research question your statement is valid because the inoculum can be relativized simply by the resuspended soil mass so as not to artificially homogenize differences that are intended to be detected from the original sample. See for examples: 10.1016/j.soilbio.2007.05.024, 10.1093/femsec/fiw178, 10.1016/j.soilbio.2016.08.004.

The same way, you must to reference this affirmation in lines 63-66, and also add to the paragraph cases where the methodological work with Ecoplates is well described according to your criteria: “Unfortunately, the studies regarding soil as a matrix for bacterial communities often include a Biolog® analysis but the details of soil preparation and dilution are often not reported, so it becomes almost impossible for other researchers to understand how to correctly use this method”. When you say that “details” are not reported, please be more specific about the methodological aspects that you, and hopefully others, would improve (a few examples would be enough).

In line 88 you state that use Tryptic Soy Agar (TSA), why this medium was chosen? It is well known that the culture has tremendous biases in representing the bacterial diversity in a soil sample (or other environmental samples), so, how the plate count with just one culture medium will realistically represent the bacteria in an inoculum?

Line 226, it is almost impossible to extract “all” the microorganisms from a soil sample.

Line 238, you cannot conclude that your protocol is useful “in a wide range of environmental conditions” if you just used a kind of sample in the experiment.

Figure S1 is not necessary, it can be easily found elsewhere.

Table S1 is very confusing and not necessary since the calculations are properly detailed in methods. Also, some researchers may prefer to make data analysis in other platforms as R. I recommend deleting Table S1.

Additional discussion about how good your methodology is compared to others that have worked with similar techniques would be of great value for the manuscript. For example, you can discuss how are SD values compared with others, or how stable is the growth curve for AWCD.

In the discussion section, the references are not indicated by numbers.

Author Response

Matera, 18 September 2019

To: The Reviewers

Applied Science

Answers to the Reviewer #1

Thank you for your suggestions and remarks. We tried to follow carefully your indications and we hope that the manuscript has been improved after the revision. In the attached files of this revised version, you will find a revised version of the manuscript in MS revision track mode and another one with the corrections already done. Here below, you will find your remarks in red and the answers to your specific questions in black. To view all the changes, please read the manuscript in MS-Word revision mode.

Comments and Suggestions for Authors

This manuscript represents a methodological contribution to an usual problem in the technical management and data analysis associated with the implementation of the BIOLOG Ecoplate test. However, I consider that it is not useful for all the possible applications of this system so you must be more precise about the standardization that you propose and for which kind of research it will be useful. The article is presented concisely, but it requires adjustments to facilitate its understanding and justify the proposal.

In this technical paper, to be eventually published in the special issue “Sustainable Agriculture and Soil Conservation”, we focused on the specific case of a sustainable-managed olive orchard, as we explained in the materials and methods section. On the other side, we have cited in the paper some articles that we published in the past about Biolog (or including Biolog analysis) in which we analyzed several types of agricultural soils at different depths. So, we surely agree that in this experiment the soils were similar but, following our protocol and calculation, everybody could find the optimal conditions (e.g., dilution, sodium pyrophosphate concentration, incubation, and so on) for specific soils. Our aim was not conducting a kind of metanalysis or writing a review, but simply writing a technical paper from an applied perspective to help other researchers overcoming the practical difficulties of this assay. We have revised the whole paper (all the corrections, replacements, rewording, new sentences and so on are highlighted in the paper in MS-Word revision mode). We hope that we managed to improve it.

You must reference this affirmation in lines 57-59: “However, the inoculated bacterial density significantly affects the rate of color development in the wells and therefore, inoculum densities have to be standardized for different soil samples using a plate count culture-based method”. It is important that you indicate for which type of research question your statement is valid because the inoculum can be relativized simply by the resuspended soil mass so as not to artificially homogenize differences that are intended to be detected from the original sample. See for examples: 10.1016/j.soilbio.2007.05.024, 10.1093/femsec/fiw178, 10.1016/j.soilbio.2016.08.004.

We agree and we admit that we were not clear. We added in the Results and Discussion that the same dilution was chosen for all the samples analyzed in the same experiment, in order to not artificially homogenize differences among the samples. We also added some references.

The same way, you must to reference this affirmation in lines 63-66, and also add to the paragraph cases where the methodological work with Ecoplates is well described according to your criteria: “Unfortunately, the studies regarding soil as a matrix for bacterial communities often include a Biolog® analysis but the details of soil preparation and dilution are often not reported, so it becomes almost impossible for other researchers to understand how to correctly use this method”. When you say that “details” are not reported, please be more specific about the methodological aspects that you, and hopefully others, would improve (a few examples would be enough).

We must admit that it is not easy comparing Biolog methodologies, as sometimes AWCD have been not calculated in the same way, sometimes papers report standard errors and not standard deviations, or vice versa, other times no SDs at all but only means, few times only OD values but not AWCD values. We tried to add some comments on these aspects in the introduction and discussion of this revised paper. But, sorry for this, we preferred to not report, for respect to other authors, all the lacks and the mistakes in calculations that we found in the literature. We have described some of them in the discussion section.

In line 88 you state that use Tryptic Soy Agar (TSA), why this medium was chosen? It is well known that the culture has tremendous biases in representing the bacterial diversity in a soil sample (or other environmental samples), so, how the plate count with just one culture medium will realistically represent the bacteria in an inoculum?

We chose TSA because is one of the most used growth media for the culturing of bacteria. TSA is a general-purpose, non-selective medium providing enough nutrients to allow for a wide variety of microorganisms to grow, and so it is suitable for enumeration of cells (counting), that is what we need. We know that just one culture medium does not represent the whole soil microflora (and neither all the possible media together, considering that, in the best case, only 10% of soil bacteria is culturable) but, in our case, the counting is indicative for the following Biolog analysis, so it does not represent the core of the experiment.

Line 226, it is almost impossible to extract “all” the microorganisms from a soil sample.

We agree and changed the text accordingly. We meant the maximum number possible of bacteria.

Line 238, you cannot conclude that your protocol is useful “in a wide range of environmental conditions” if you just used a kind of sample in the experiment.

We agree and changed the text accordingly, as you can see in the revised paper (revision mode).

Figure S1 is not necessary, it can be easily found elsewhere.

Table S1 is very confusing and not necessary since the calculations are properly detailed in methods. Also, some researchers may prefer to make data analysis in other platforms as R. I recommend deleting Table S1.

We agree regarding Fig. S1. Table S1 was also deleted and some details on the calculations added.

Additional discussion about how good your methodology is compared to others that have worked with similar techniques would be of great value for the manuscript. For example, you can discuss how are SD values compared with others, or how stable is the growth curve for AWCD.

As reported above, we have added in the discussion some comments on the main Biolog issues, especially regarding inoculum density. We also commented and compared our results with those of other authors in reference to SDs of the AWCD values and AWCD curve stability.

In the discussion section, the references are not indicated by numbers.

We corrected the references without numbers.

Best regards,

Adriano Sofo

Professor

Reviewer 2 Report

Overview:

This technical note attempts to suggest a method for standardizing the use of the Biolog Ecoplates assay. I believe they have done so within a single experiment where all the soils are generally similar. I am not convinced that the proposed parameters are universal to other soils. I believe the text needs to make it explicit that what the authors are suggesting is to perform this method comparison during each experiment, i.e. determining the AWCD leveling off point at incubation time and Na4P2O7 • 10H2O concentration for each soil sample. Or, the authors should demonstrate that their parameters are robust to different soil types. Lastly, the authors mention diversity in the title and methods but do not present any diversity data. I think a comparison of how the diversity metrics differ under different inoculation density and buffer concentration are necessary to show that this method is needed. If there are no differences in the results that other researchers would be presenting then I am not sure that the standardization is necessary. Otherwise, I appreciate the authors diligence in presenting the data related to determining parameters that can be used to standardize this assay and I believe that with some modifications to the manuscript that this will be a useful addition for microbial ecologists.

Abstract

Line 29: “researches” should be “research”

Line 34-35: I think the authors intend to use the word conservation and instead use “conservative.” This word usage occurs throughout the manuscript.

Introduction

The introduction clearly lays out the problems with the Biolog assay, namely that standardization is lacking, and proposes that standardization would make this assay easier to interpret within and between experiments. I think that overall design of the introduction is good for the manuscript. The one thing I am not clear on is whether or not there is whether anyone has tried standardizing it in the past. Below are some suggestions for improving the clarity of a few points.

Line 40: I suggest rewording this to “present in virtually all ecosystems” otherwise the implication is that microorganisms are virtual.

Line 68-69. Either “a soil bacterial community…” or “soil bacterial communities”

Line 69: What is meant by “symbols” in this context? I think this should be more explicit, because at this point in the paper I have no idea what sort of symbol the authors are developing. Generally, I would think of a symbol as being some sort of graphical representation, but I do not think that is what is meant here.

Line 73-76. I think this sentence could be simplified to something like “The standardization of this technique could be essential for indicating differences in CLPPs between samples within a single experiment, for comparing different experiments, soil types and management systems, or simply soils sampled at different times.” This isn’t something I think has to change, but this sentence is overly complicated for expressing a fairly simple idea.

Materials and Methods

The methods appear to address the primary issue of standardizing this procedure. However, there were a few areas that I found unclear. I have two questions I think should be addressed in some way in the methods. 1) The authors mention that the there is no clear definition of a blank in their introduction, yet they only mention the blank as a way to determine incubation time. I know that they later explain how to correctly calculate a blank in the results but the methods should explicitly include an explanation that this will be done. 2) Why not use some sort of beta diversity method to visualize and determine differences in the suite of C sources, such as PCA/PCOA and PERMANOVA?

Line 97-98: I suggest, “soil particles were allowed to settle at…”

Line 99: suggest substituting “performed” for “done”

Line 101: I suggest using exponent notation, or substitute “per” for “for”

Line 102: I suggest using “~” instead of “ca.”

Line 104-16: I believe this section should include the agitator

Line 107-109: This is unclear, are the authors suggesting that the same tube that was used for the serial dilutions be used for the assay 72+ hours after incubating the plates? If so, I believe there needs to be some evidence provided that storage in the buffer/refrigerator doesn’t alter viable cell counts as well as confirmation that the CLPP doesn’t change during that time frame. If the authors used a new dilution then that should be made clear here. Also the last word should be “microplate.”

Line 113: The authors should give an idea of what “daily” means here, e.g. daily (12 h +/- X  h). I chose 12 since the authors report at a 12 hour increment.

Line 149: It is unclear what is meant by “indications” here.

Line 165: should be “crop” singular

Line 173-180: Were these soils sieved or homogenized?

Results and Discussion

The results clearly relate the findings of this study. However, this section would benefit from a couple of minor edits, as well as the figures being clearly referred to in the text. I am also confused as to why the methods mention diversity metrics but none of those data are presented here.

Line 183-194: I am not sure if I am missing something, but it seems that all the references to figure 2 should be references to figure 1.

Line 221-222: should read “micropore and macropore walls”

Figure 1: Why are results only shown to 144 h but the methods mention up to 167 h. I suggest displaying the full 167 h data, explaining why it is not included, or modifying the methods to reflect what was actually done.

Figure 2: I am not sure why there is a line connecting the values. Were these discreet vials for each measurement? If so I don’t think the line should be in this figure.

Conclusion

The conclusion would benefit from summarizing the important parts of the study for each of the 4 points. E,g, 10^5 cfu/ml etc. Also, I am still not sure what an unequivocal symbol is in this usage.

Line 240-241: see above for use of “conservative”

Author Response

Matera, 18 September 2019

To: The Reviewers

Applied Science

Answers to the Reviewer #2

Thank you for your suggestions and remarks. We tried to follow carefully your indications and we hope that the manuscript has been improved after the revision. In the attached files of this revised version, you will find a revised version of the manuscript in MS revision track mode and another one with the corrections already done. Here below, you will find your remarks in red and the answers to your specific questions in black. To view all the changes, please read the manuscript in MS-Word revision mode.

Comments and Suggestions for Authors

Overview:

This technical note attempts to suggest a method for standardizing the use of the Biolog Ecoplates assay. I believe they have done so within a single experiment where all the soils are generally similar. I am not convinced that the proposed parameters are universal to other soils. I believe the text needs to make it explicit that what the authors are suggesting is to perform this method comparison during each experiment, i.e. determining the AWCD leveling off point at incubation time and Na4P2O7 • 10 H2O concentration for each soil sample. Or, the authors should demonstrate that their parameters are robust to different soil types.

In this technical paper, to be eventually published in the special issue “Sustainable Agriculture and Soil Conservation”, we focused on the specific case of a sustainable-managed olive orchard, as we explained in the materials and methods section. On the other side, we have cited in the paper some articles that we published in the past about Biolog (or including Biolog analysis) in which we analyzed several types of agricultural soils at different depths. So, we surely agree that in this experiment the soils were similar but, following our protocol and calculation, everybody could find the optimal conditions (e.g., dilution, sodium pyrophosphate concentration, incubation, and so on) for specific soils. Our aim was not conducting a kind of metanalysis or writing a review, but simply writing a technical paper from an applied perspective to help other researchers overcoming the practical difficulties of this assay.

Lastly, the authors mention diversity in the title and methods but do not present any diversity data. I think a comparison of how the diversity metrics differ under different inoculation density and buffer concentration are necessary to show that this method is needed. If there are no differences in the results that other researchers would be presenting then I am not sure that the standardization is necessary. Otherwise, I appreciate the authors diligence in presenting the data related to determining parameters that can be used to standardize this assay and I believe that with some modifications to the manuscript that this will be a useful addition for microbial ecologists.

In this revised version, we presented data on diversity indices (H’ and E) in two new figures (Fig. 2 and Fig. 3), together with AWCD values.

Abstract

Line 29: “researches” should be “research”

Line 34-35: I think the authors intend to use the word conservation and instead use “conservative.” This word usage occurs throughout the manuscript.

Done.

Introduction

The introduction clearly lays out the problems with the Biolog assay, namely that standardization is lacking, and proposes that standardization would make this assay easier to interpret within and between experiments. I think that overall design of the introduction is good for the manuscript. The one thing I am not clear on is whether or not there is whether anyone has tried standardizing it in the past.

There are many papers on this technique and a critique on this method was moved by Preston-Mafham, J.; Boddy, L.; Randerson, P.F. (Analysis of microbial community functional diversity using sole-carbon-source utilisation profiles: a critique. FEMS Microbiol. Ecol. 2002, 42, 1–14) but, from what we know, nobody tried to standardize it before.

Below are some suggestions for improving the clarity of a few points.

Line 40: I suggest rewording this to “present in virtually all ecosystems” otherwise the implication is that microorganisms are virtual.

Done.

Line 68-69. Either “a soil bacterial community…” or “soil bacterial communities”

Done.

Line 69: What is meant by “symbols” in this context? I think this should be more explicit, because at this point in the paper I have no idea what sort of symbol the authors are developing. Generally, I would think of a symbol as being some sort of graphical representation, but I do not think that is what is meant here.

We meant terminology and parameters more than symbols. We corrected this in the text.

Line 73-76. I think this sentence could be simplified to something like “The standardization of this technique could be essential for indicating differences in CLPPs between samples within a single experiment, for comparing different experiments, soil types and management systems, or simply soils sampled at different times.” This isn’t something I think has to change, but this sentence is overly complicated for expressing a fairly simple idea.

Done.

Materials and Methods

The methods appear to address the primary issue of standardizing this procedure. However, there were a few areas that I found unclear. I have two questions I think should be addressed in some way in the methods. 1) The authors mention that the there is no clear definition of a blank in their introduction, yet they only mention the blank as a way to determine incubation time. I know that they later explain how to correctly calculate a blank in the results but the methods should explicitly include an explanation that this will be done.

We have corrected this issue.

2) Why not use some sort of beta diversity method to visualize and determine differences in the suite of C sources, such as PCA/PCOA and PERMANOVA?

For clarity, we decided to convert the Table 1 in Figure 1. Now it should be visually clear that AWCD changes according to the calculation. This obviously also caused changes in H’ and E, that indeed in many papers are erroneously calculated.

Line 97-98: I suggest, “soil particles were allowed to settle at…”

Line 99: suggest substituting “performed” for “done”

Line 101: I suggest using exponent notation, or substitute “per” for “for”

Line 102: I suggest using “~” instead of “ca.”

Line 104-106: I believe this section should include the agitator

Done.

Line 107-109: This is unclear, are the authors suggesting that the same tube that was used for the serial dilutions be used for the assay 72+ hours after incubating the plates? If so, I believe there needs to be some evidence provided that storage in the buffer/refrigerator doesn’t alter viable cell counts as well as confirmation that the CLPP doesn’t change during that time frame. If the authors used a new dilution then that should be made clear here.

The plate counting is carried out in parallel with the Biolog, using different solutions. Otherwise there would be the problem you say, but it was not our case.

Also the last word should be “microplate.”

Done.

Line 113: The authors should give an idea of what “daily” means here, e.g. daily (12 h +/- X h). I chose 12 since the authors report at a 12 hour increment.

We change this in the whole manuscript.

Line 149: It is unclear what is meant by “indications” here.

This paragraph, according to the suggestions of Reviewer #1, has been removed.

Line 165: should be “crop” singular

Done.

Line 173-180: Were these soils sieved or homogenized?

Visible crop residues, roots and pebbles were removed with sterile tweezers, and the soil were slightly mixed and homogenized with a sterile spatula. We analyzed fresh soil, so it was not possible to sieve it. For avoiding contamination, it would be better not to manipulate too much the soil before Biolog analysis.

Results and Discussion

The results clearly relate the findings of this study. However, this section would benefit from a couple of minor edits, as well as the figures being clearly referred to in the text. I am also confused as to why the methods mention diversity metrics but none of those data are presented here.

In this revision version, we presented data on diversity indices (H’ and E) in two new figures (Fig. 2 and Fig. 3), together with AWCD values.

Line 183-194: I am not sure if I am missing something, but it seems that all the references to figure 2 should be references to figure 1.

Corrected.

Line 221-222: should read “micropore and macropore walls”

Corrected.

Figure 1: Why are results only shown to 144 h but the methods mention up to 167 h. I suggest displaying the full 167 h data, explaining why it is not included, or modifying the methods to reflect what was actually done.

It was a mistake, sorry. It was up to 144 h and not 167 h. We corrected it.

Figure 2: I am not sure why there is a line connecting the values. Were these discreet vials for each measurement? If so I don’t think the line should be in this figure.

We removed the line. Moreover, the new Fig. 2 now includes H’ and E data, as requested.

Conclusion

The conclusion would benefit from summarizing the important parts of the study for each of the 4 points. E,g, 10^5 cfu/ml etc. Also, I am still not sure what an unequivocal symbol is in this usage.

We corrected the discussion, specifying the main results.

Line 240-241: see above for use of “conservative”

Done.

Best regards,

Adriano Sofo

Professor

Round 2

Reviewer 1 Report

Thanks for the corrections made, the work improved substantially since now it is clearly stated that the protocol is proposed for a particular kind of research question and the conclusions are made in consequence with that.
If the manuscript will be reviewed by English Editors, I have no problem if it is accepted in the current state.